# Epigenetic and transcriptional dysregulation in CD4+ T cells in patients with atopic dermatitis

Amy A. Eapen[1,2,3☯], Sreeja Parameswaran[3☯], Carmy Forney[3], Lee E. Edsall[3], Daniel Miller[3], Omer Donmez[3], Katelyn Dunn[3], Xiaoming Lu[3], Marissa Granitto[3], Hope Rowden[3], Adam Z. Magier[1], Mario Pujato[3], Xiaoting Chen[3], Kenneth Kaufman[3,4,5,6], David I. Bernstein[7], Ashley L. Devonshire[1,5], Marc E. Rothenberg[1,5], Matthew T. Weirauch[3,4,5]*, Leah C. Kottyan[1,3,5]*

1 Division of Allergy and Immunology, Cincinnati Children's Hospital Medical Center, Cincinnati, Ohio, United States of America, 2 Division of Allergy and Clinical Immunology, Henry Ford Health System, Detroit, Michigan, United States of America, 3 Center for Autoimmune Genomics and Etiology, Cincinnati Children's Hospital Medical Center, Cincinnati, Ohio, United States of America, 4 Divisions of Biomedical Informatics and Developmental Biology, Cincinnati Children's Hospital Medical Center, Cincinnati, Ohio, United States of America, 5 Department of Pediatrics, University of Cincinnati College of Medicine, Cincinnati, Ohio, United States of America, 6 Cincinnati Veterans Administration, Cincinnati, Ohio, United States of America, 7 Division of Immunology, Allergy, and Rheumatology, University of Cincinnati, College of Medicine, Cincinnati, Ohio, United States of America

☯ These authors contributed equally to this work.
* Matthew.Weirauch@cchmc.org (MTW); Leah.Kottyan@cchmc.org (LCK)

**Data Availability Statement:** All raw and processed sequencing data generated in this study have been submitted to the NCBI Gene Expression Omnibus (GEO) (https://www.ncbi.nlm.nih.gov/

## Abstract

Atopic dermatitis (AD) is one of the most common skin disorders among children. Disease etiology involves genetic and environmental factors, with 29 independent AD risk loci enriched for risk allele-dependent gene expression in the skin and CD4+ T cell compartments. We investigated the potential epigenetic mechanisms responsible for the genetic susceptibility of CD4+ T cells. To understand the differences in gene regulatory activity in peripheral blood T cells in AD, we measured chromatin accessibility (an assay based on transposase-accessible chromatin sequencing, ATAC-seq), nuclear factor kappa B subunit 1 (NFKB1) binding (chromatin immunoprecipitation with sequencing, ChIP-seq), and gene expression levels (RNA-seq) in stimulated CD4+ T cells from subjects with active moderate-to-severe AD, as well as in age-matched non-allergic controls. Open chromatin regions in stimulated CD4+ T cells were highly enriched for AD genetic risk variants, with almost half of the AD risk loci overlapping AD-dependent ATAC-seq peaks. AD-specific open chromatin regions were strongly enriched for NF-κB DNA-binding motifs. ChIP-seq identified hundreds of NFKB1-occupied genomic loci that were AD- or control-specific. As expected, the AD-specific ChIP-seq peaks were strongly enriched for NF-κB DNA-binding motifs. Surprisingly, control-specific NFKB1 ChIP-seq peaks were not enriched for NFKB1 motifs, but instead contained motifs for other classes of human transcription factors, suggesting a mechanism involving altered indirect NFKB1 binding. Using DNA sequencing data, we identified 63 instances of altered genotype-dependent chromatin accessibility at 36 AD risk variant loci (30% of AD risk loci) that might lead to genotype-dependent gene expression. Based on these findings, we propose that CD4+ T cells respond to stimulation in an AD-specific

geo/) database under accession number GSE184238. A UCSC Genome Browser session is available at http://genome.ucsc.edu/s/ledsall/AtopicDermatitis.

**Funding:** This study was funded in part by National Institutes of Health (NIH) through R01 DK107502, R01 AI148276, U19 AI070235, R01 AI130830, U01 HG011172, and P30 AR070549 to LCK. NIH funding awarded to MW that contributed to this study includes R01 HG010730, R01 NS099068, R01 GM055479, and U01 AI130830. NIH awards R01 AR073228 and R01 AI024717 to MW and LCK also contributed to this study. The Cincinnati Children's Hospital Medical Center ARC Award 53632 to MW and LCK further contributed to this study. The funders had no role in study design, data collection and analysis, decision to publish, or preparation of the manuscript.

**Competing interests:** The authors have declared that no competing interests exist.

manner, resulting in disease- and genotype-dependent chromatin accessibility alterations involving NFKB1 binding.

## Author summary

Gene expression is regulated in stimulated CD4+ T cells in a disease-dependent manner in patients with atopic dermatitis (AD). AD-specific regions of chromatin accessibility and binding of the NFKB1 transcription factor are enriched for AD genetic risk variants. Clinically, CD4+ T cells in the peripheral blood of patients with AD respond to stimulation in a disease- and genotype-dependent manner.

## Introduction

Atopic dermatitis (AD) is one of the most common skin disorders in children, affecting nearly 20% of children worldwide and constituting a significant social and financial burden for affected children and their families [1]. Although AD onset often occurs during childhood, up to 80% of patients with AD have persistent disease continuing into adulthood [2,3]. Currently, patients with moderate-to-severe AD are treated using a "one-size-fits-all" approach, but recent investigations have revealed several different AD endotypes [4]. Both genetic and environmental factors have been implicated in AD pathogenesis [5], and 29 independent AD risk loci have been identified using genome-wide association studies (GWAS) [6,7].

Immunologically, AD involves skin barrier defects and CD4+ T cells that localize to the skin, producing inflammatory cytokines and amplifying epidermal dysfunctions [8]. This can result in allergic sensitization via disruption of the skin barrier, contributing to subsequent development of other allergic diseases along the atopic march, including allergic rhinitis, food allergies, and asthma [9]. Recent studies suggest that early and aggressive management of AD may prevent allergic sensitization and further progression of the atopic march [10–12].

AD genetic risk variants are enriched for genes with genotype-dependent expression (expression quantitative trait loci, eQTLs) in the skin and in CD4+ T cells. This study focused on CD4+ T cells because of their critical role in shaping the immune response in AD and other allergic diseases. Notably, in transcriptional studies on food allergies, the most robust disease-specific expression in CD4+ T cells was detected after stimulation with CD3/CD28 or antigen-loaded dendritic cells [13–15], the two stimulatory pathways that activate the nuclear factor kappa B (NF-κB) protein [16,17]. NF-κB signaling controls the transcription of inflammatory cytokines, such as interleukin-6, and adhesion molecules such as the intercellular adhesion molecule-1, which is responsible for skin inflammation and disruption of the skin barrier [18,19]. An important role for NF-κB in CD4+ T cells in AD was recently established using a mouse model of AD, in which mice injected with CD4+ T cells with inhibited NF-κB signaling showed marked improvement in AD-like skin lesions compared to those injected with CD4+ T cells with a control vector [20].

In this study, we hypothesized that that the chromatin accessibility and binding of transcription factors may regulate genetic AD risk loci. To test this hypothesis, we measured chromatin accessibility, NFKB1 binding, and gene expression levels in stimulated CD4+ T cells in subjects with active moderate-to-severe AD as well as in age- and ancestry-matched healthy and non-allergic controls. We identified 34,216 chromatin regions across the genome that were accessible in an AD-dependent manner. These regions were highly enriched for DNA sequence motifs recognized by NF-κB transcription factors (TFs). We performed ChIP-seq for

**Table 1. Expression quantitative trait loci (eQTL) enrichment at atopic dermatitis (AD) risk loci.** Application of our regulatory element locus intersection (RELI) method [21] to 3,143 AD variants across 29 independent risk loci using eQTL data obtained from the Genotype-Tissue Expression (GTEx) database [22] and the Database of Immune Cell Expression, eQTL and Epigenomics (DICE) [23].

| Cell line/track | Overlap | Corrected P-value | Enrichment |
|---|---|---|---|
| CD4$^+$ T cell: DICE | 6/29 | $3.85 \times 10^{-5}$ | 6.84 |
| Sun-unexposed skin: GTEx | 14/29 | $4.91 \times 10^{-8}$ | 4.67 |
| Sun-exposed skin: GTEx | 10/29 | $2.9 \times 10^{-4}$ | 4.19 |
| Esophageal mucosa: GTEx | 8/29 | $2.34 \times 10^{-6}$ | 4.09 |

NFKB1 in patients with AD and in control individuals, and identified 20,322 genomic loci with AD-dependent NFKB1 occupancy. Whole genome sequencing of these individuals and the application of our Measurement of Allelic Ratios Informatics Operator (MARIO) method [21] revealed 63 instances of genotype-dependent chromatin accessibility at 36 AD risk variant loci that might lead to genotype-dependent gene expression. Collectively, our findings demonstrate that the pathoetiology of AD involves epigenetic changes in CD4$^+$ T cells via NF-κB-mediated gene expression regulation.

## Results

We created a set of 3,143 AD-associated genetic risk variants at 29 independent risk loci (See **METHODS** and **S1 Table**). Application of our Regulatory Element Locus Intersection (RELI) method [21] to these variants using expression quantitative trait loci (eQTL) data obtained from the Genotype-Tissue Expression (GTEx) database [22] and the Database of Immune Cell Expression, eQTL, and Epigenomics (DICE) [23] revealed the strongest enrichment for CD4$^+$ T cells and in the skin (sun-exposed and sun-unexposed) (**Table 1**). This analysis indicates that alteration of gene regulatory mechanisms in CD4$^+$ T cells is likely an important factor underlying AD-associated genetic risk.

We recruited six patients with moderate-to-severe AD (average eczema area and severity index (EASI) score = 30) and six age- and ancestry-matched controls. Participant demographics are presented in **Table 2**. Adults with persistent AD had experienced disease onset in childhood. As expected, the mean total IgE levels in AD subjects (180.8 kU/L) were higher than those in the controls (61.7 kU/L). Peripheral blood was obtained from each subject and CD4$^+$ T cells were isolated and stimulated for 45 h with anti-CD3/CD28 beads (**Fig 1**).

### Global mapping of the chromatin accessibility landscape in AD CD4$^+$ T cells

We performed assay for Transposase-Accessible Chromatin followed by sequencing (ATAC-seq) to map genome-wide chromatin accessibility. The data obtained were of high quality,

**Table 2. Demographics of six age-matched AD case and control subjects.**

| | Age (years) | Gender (% male) | EASI (0–72) | IGA (1–4) | Age of onset (years) | Asthma | Allergic rhinitis | Food allergy | Total IgE | Number of sensitizations |
|---|---|---|---|---|---|---|---|---|---|---|
| Children with active AD (n = 2) | 10 (8–12) | 1/2 (50%) | 31.9 (31.3–32.4) | 3.5 (3–4) | 4.5 (2–7) | 0 | 1/2 (50%) | 0 | 16.5 (6–27) | 3.5 (1–6) |
| Children without AD (n = 2) | 13.5 (10–17) | 1/3 (33%) | n/a | n/a | n/a | n/a | n/a | n/a | 150.5 (23–278) | n/a |
| Adults with active AD (n = 4) | 44.2 (25–67) | 1/4 (25%) | 29.2 (17.1–50.9) | 3.3 (3–4) | 4 (0.33–10) | 3/4 (75%) | 5/5 (100%) | 3/4 (75%) | 290.3 (18–648) | 5.3 (4–8) |
| Adults without AD (n = 4) | 35.3 (18–53) | 0/4 (0%) | n/a | n/a | n/a | n/a | n/a | n/a | 17.2 (8–36) | n/a |

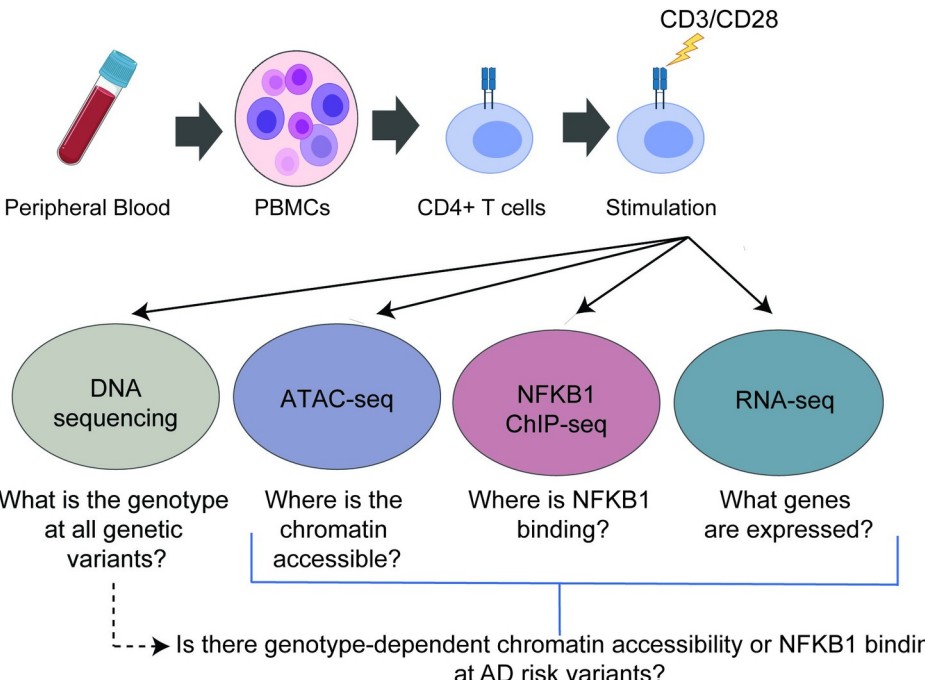

**Fig 1. Study Design.** Figure created in BioRender.

with an average of almost 70,000 peaks per dataset, an average fraction of reads inside peaks (FRiP) score of 0.32, and an average transcription start site (TSS) enrichment score of 20.5 (**S2 Table**). Pairwise comparisons of each dataset identified strong agreement between subjects within case and control cohorts (**S1A and S1B Fig**).

We assessed the overlap of chromatin accessibility data with AD genetic risk variants using the RELI method [21]. Seven of the twelve ATAC-seq datasets were significantly enriched for AD risk loci with 7–14 overlapped risk loci for each subject (RELI $p_{corrected}$ = 0.01–1.6 x $10^{-3}$) (**S3 Table**).

In pairwise assessments performed using MAnorm [24], most ATAC-seq peaks were shared between AD patients and demographically matched controls (85.9–96.0%). The remaining peaks were either stronger in AD (AD-specific) or control (control-specific) groups (representative analysis in **Fig 2A**; full results in **S2 Fig**). We identified 34,216 regions of chromatin across the genome that were accessible in an AD-dependent manner, yielding 357 AD-specific and 343 control-specific peaks that were present in three or more pairs (**S3 Fig**). We defined these ATAC-seq peaks that were AD-specific or control-specific in three or more subject pairs as "consistently AD-specific" and "consistently control-specific" peaks, respectively. Consistently AD-specific ATAC-seq peaks overlapped AD-associated genetic risk variants at 13 of the 29 AD risk loci (2.0-fold enrichment, $p_{corrected}$ = 0.015) (**S3 Table**). As expected, there was substantial overlap between AD-consistent and control-specific ATAC-seq peaks since most chromatin states do not change from individual to individual within a cell type. We identified 5,631 AD-specific consistent ATAC-seq peaks (i.e., consistent peaks that were not consistent in controls). Likewise, we identified 10,058 control-specific consistent ATAC-seq peaks. In total, we identified 357 consistently AD-specific ATAC-seq peaks. Consistently AD-specific ATAC peaks have no overlap consistently control-specific ATAC peaks. These results indicate that chromatin is accessible in a disease-specific manner in CD4$^+$ T cells at almost half of the AD risk loci.

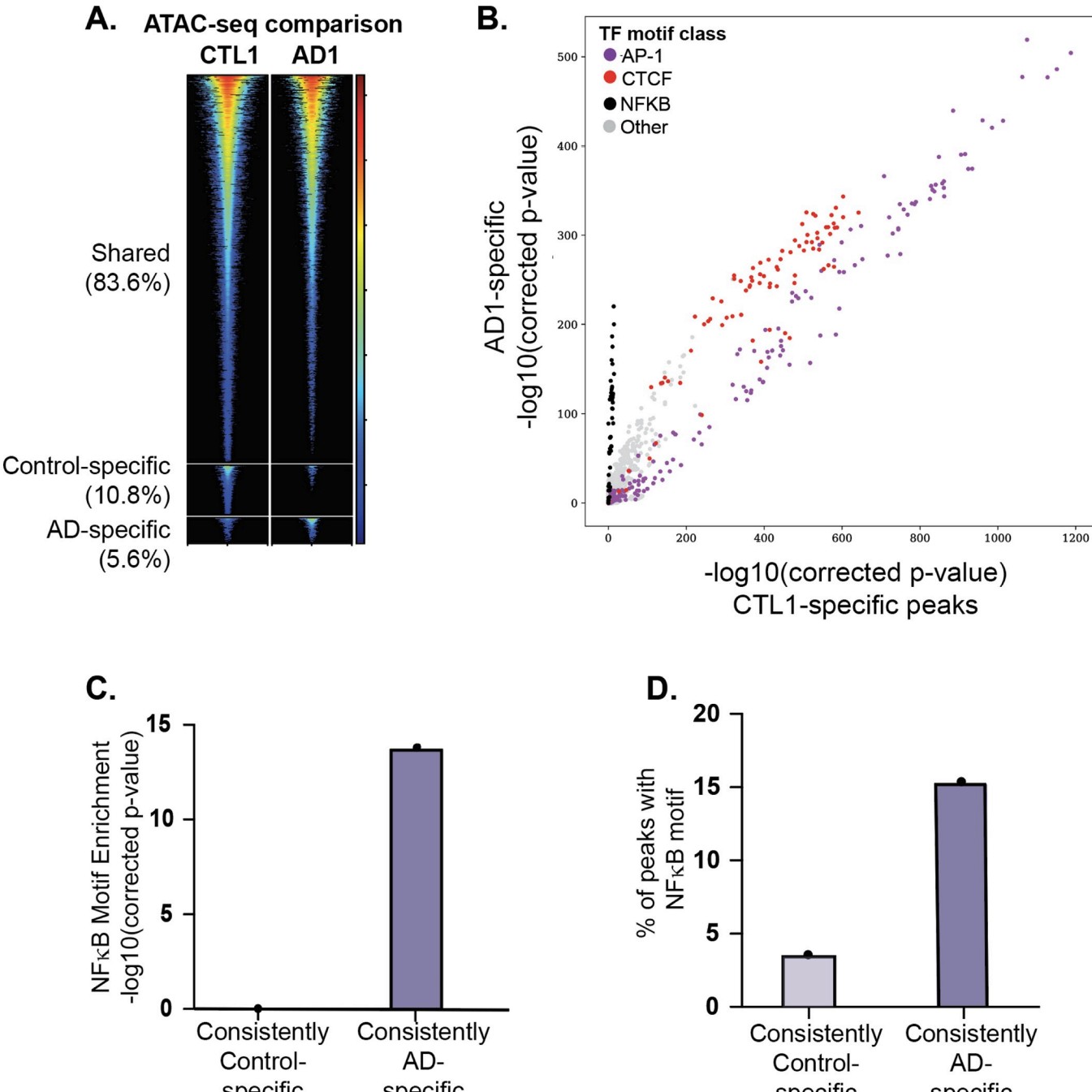

**Fig 2. Differential chromatin accessibility and transcription factor (TF) motif enrichment in atopic dermatitis (AD) subjects vs. matched controls.** Assay for transposase-accessible chromatin sequencing (ATAC-seq) peaks were identified for all cases and controls and compared for all subject pairs. (A) Differential chromatin accessibility analysis. For each matched pair of subjects, we identified shared, control-specific, and AD-specific peaks (see METHODS). A representative subject pair is shown in (A). Each row represents a single genomic locus where an ATAC-seq peak was identified in either the AD or control subject. The center of each row corresponds to the center of the ATAC-seq peak. Heatmap colors indicate the normalized ATAC-seq read count within the AD1 (right) or CTL1 (control) (left) subject–see key on the right. (B) Differential transcriptional factor (TF) motif enrichment analysis. Comparison of TF motif enrichment results within a representative AD-specific and control-specific matched subject pair. Each dot represents the enrichment of a particular motif (corrected negative $\log_{10}$ p-value). Select motif families are color-coded (see key on the upper left side). (C and D) Nuclear factor kappa B (NFKB) motif enrichment comparison between consistently AD-specific and consistently control-specific ATAC-seq peaks. "Consistently specific" peaks were defined as those peaks that were AD- or control-specific in at least three AD or control subjects, respectively. Results are shown for representative Cis-BP NFKB motif, M05887_2.00. Full results are provided in **S5 Table**.

To identify potential TFs whose binding might be affected by differential chromatin accessibility, we performed TF binding site motif enrichment analysis on AD- and control-specific ATAC-seq peaks. These analyses revealed that NF-κB DNA-binding motifs were more strongly enriched in the AD-specific ATAC-seq peaks than in the control-specific ATAC-seq peaks in five of the six matched pairs, with the remaining pair showing equivalent enrichment for NF-κB (**Figs 2B**, **S2**, **and S4 Table**). Motif enrichment analysis in consistently AD- and control-specific peaks confirmed that NF-κB binding motifs were highly enriched in an AD-specific manner, with ~17% of consistently AD-specific peaks containing predicted NFKB binding sites ($P < 10^{-13}$) compared to ~4% of consistently control-specific peaks ($P = 1$) (**Fig 2C and 2D** **and S5 Table**). These data highlight the potential for more robust direct binding of NFKB to the genome in activated CD4$^+$ T cells in the AD patients than in the matched controls.

## NFKB1 binds in an AD-specific manner at hundreds of genomic loci in CD4$^+$ T cells

We performed NFKB1 (p50) chromatin immunoprecipitation with sequencing (ChIP-seq) experiments to measure NF-κB binding to the genome within stimulated CD4$^+$ T cells, and obtained an average of approximately 11,000 peaks per subject, with an average FRiP score of 0.012 (**S2 Table**). All ChIP-seq peak datasets displayed a highly significant overlap with a previously published CD4$^+$ T cell NFKB1 ChIP-seq dataset (GSE126505) (**S6 Table**). As expected, the NF-κB DNA-binding motif was highly enriched in each of our NF-κB ChIP-seq datasets (Cis-BP NF-κB motif M05887_2.00 enrichment: $10^{-4158} < P < 10^{-123}$ (**S7 Table**)). We identified 20,322 genomic loci with AD-dependent NFKB1 occupancy. AD-specific NFKB1 ChIP-seq peaks were enriched for overlap with AD-specific ATAC-seq peaks in all six pairs (between 5.9 and 38.1-fold enrichment, $3.5 \times 10^{-25} < P < 3.20 \times 10^{-203}$) (**S8 Table**, reporting AD-specific ATAC and AD-specific ChIP). A global variability analysis of ATAC and ChIP peaks revealed minimal differences in variability within cases versus within controls. Substantially more variability was observed between subject matched pairs in the NFKB1 ChIP-seq experiments than in the ATAC-seq experiments, with a median of 51.5% shared NFKB1 peak as compared to a median of 91.9% shared ATAC-seq peaks (**Fig 3**). These results indicate substantially more differential NFKB1 binding than chromatin accessibility in AD subjects compared to that observed in matched controls.

wWe next sought to identify AD- and control-specific NFKB1 binding events by performing a pairwise assessment of NFKB1 peaks in the case vs. control subjects (see METHODS). This procedure identified shared, control-specific, and AD-specific NFKB1 peaks. A representative pair is shown in **Fig 4A**, and all pairs are shown in **S4 Fig**. Strikingly, NF-κB binding sites were more strongly enriched in the AD-specific NFKB1 ChIP-seq peaks than in the matched controls in five pairs (**S4 Fig** and **S7 Table** (a representative pair is shown in **Fig 4B**). We defined NFKB1 peaks that were AD-specific and/or control-specific in three or more subject pairs as "consistently AD-specific" and "consistently control-specific" peaks, respectively (**S3C and S3D Fig**). In total, we identified 131 AD-specific and 72 control-specific NFKB1 ChIP-seq peaks. Motif enrichment analysis revealed that NFKB binding motifs were also the most highly enriched motif class within the consistently AD-specific NFKB1 peaks (**Fig 4C, 4D** and **S9 Table**). In contrast, consistently control-specific peaks were not enriched for NFKB motifs, but instead were enriched for a wide range of other motif classes (**S5A and S5B Fig** and **S9 Table**). Collectively, these results indicate that AD-specific NFKB ChIP-seq peaks are strongly enriched for NFKB1 motifs, whereas control-specific NFKB peaks surprisingly are not.

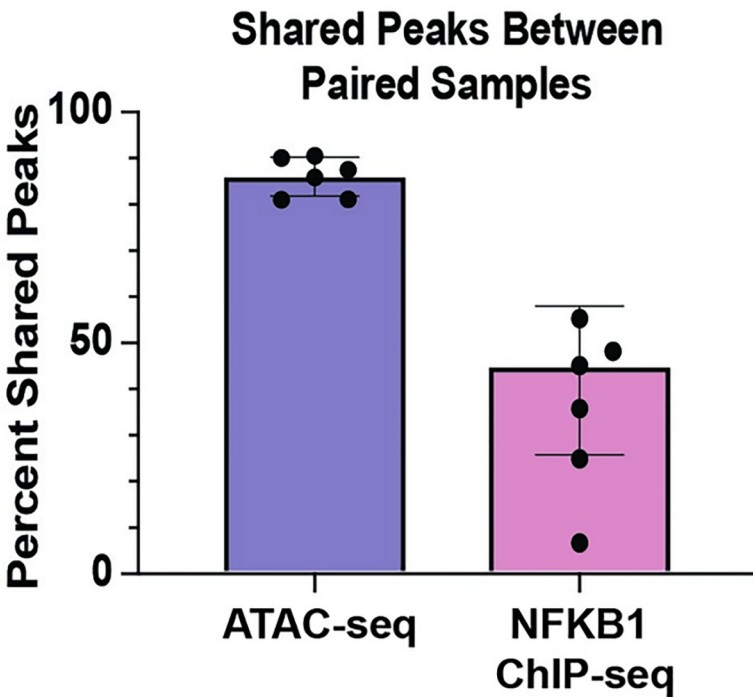

**Fig 3. Shared peaks in ATAC-seq and chromatin immunoprecipitation with sequencing (ChIP-seq) experiments between paired subjects.** Percentage of shared peaks between paired samples in ATAC-seq compared to NFKB1 ChIP-seq experiments.

## Comparisons across different data types confirm the strong concordance among RNA-, ATAC-, and NFKB1 ChIP-seq results

Next, we measured gene expression levels in CD3/CD28-stimulated CD4$^+$ T cells in each subject, with the goal of integrating these data with the chromatin accessibility and NFKB1 binding data. In case-control pairwise analysis, 15 genes were expressed at least 1.5-fold higher in the stimulated CD4$^+$ T cells from patients with AD compared to the matched controls, while 16 genes were expressed 1.5-fold lower in the case group compared to the matched control group (**S10 Table**). These 31 genes were enriched for AD-related processes, such as the "regulation of immune system processes", "lymphocyte activation", and "cytokine-mediated signaling pathway" gene ontology (GO) biological processes as well as the "cytokine receptor binding", "nitric oxide synthase binding", and "RNA-polymerase II-specific DNA-binding transcription factor binding" GO molecular functions (**S6 Fig**).

The 100 kB region of DNA around AD-specific gene sets widely overlapped (94.7–100%) the ATAC-seq peaks in the six subjects with AD (**S8 Table**). There was substantial overlap (26.3–68.4%) between the 100 kb region of DNA around AD-specific gene sets with the AD-specific ATAC-peaks (**Fig 5**), indicating that possible enhancers proximal to the AD-specific genes were accessible for transcription in an AD-specific manner. Similarly, the 100 kb region of DNA around AD-specific NFKB1 ChIP-seq peaks overlapped the transcriptional start site of 47–95% of the AD-specific genes (**S8 Table** and **Fig 5**). In five of the six pairs, AD-specific NFKB1 ChIP-seq peaks overlapped a large proportion of the AD-specific genes (42.1–73.4%) (**S8 Table**). The 100 kb region around NFKB1 ChIP-seq peaks specific in AD overlapped 44.3% of the genes with increased expression in AD compared to the 15.4% of genes with increased expression in AD that were overlapped by 100 kb region around NFKB1 ChIP-seq

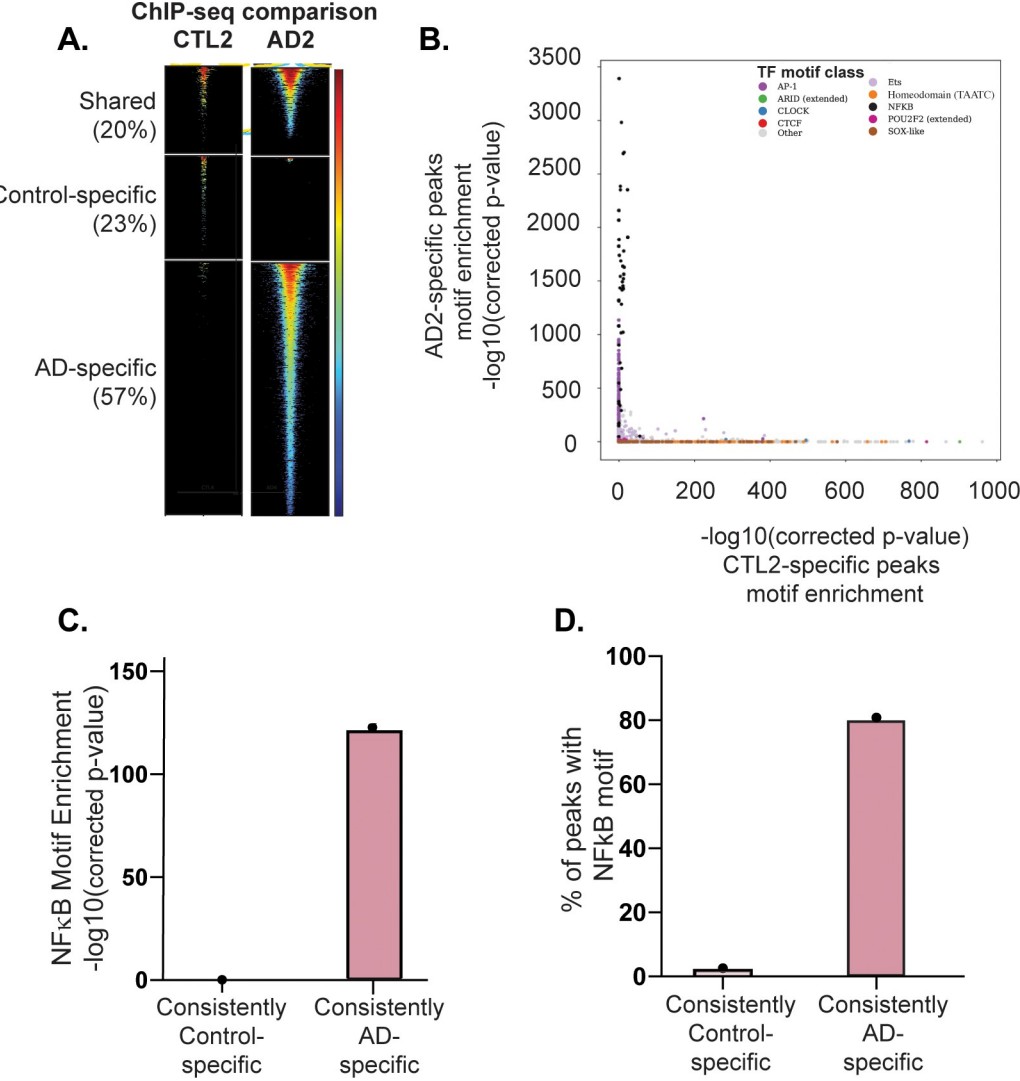

**Fig 4. Differential NFKB1 binding and TF motif enrichment in AD subjects vs. matched controls.** NFKB1 ChIP-seq peaks were identified for all case and control subjects and compared between matched subject pairs. (A) Differential NFKB1 binding analysis. For each matched pair of subjects, we identified the shared peaks, control-specific peaks, and AD-specific peaks (see METHODS). A representative subject pair is shown in (A). Each row represents a single genomic locus, where an NFKB1 ChIP-seq peak was identified in either the AD or control subject. The center of each row corresponds to the center of the ChIP-seq peak. Heatmap colors indicate the normalized ChIP-seq read counts within the AD2 (right) or CTL2 (left) subject–see key on the right. (B) Differential TF motif enrichment analysis. Comparison of TF motif enrichment results within a representative AD-specific and control-specific matched subject pair. Each dot represents the enrichment of a particular motif (corrected negative log$_{10}$ p-value). Select motif families are color-coded (see key on the upper right side). (C and D) NFKB motif enrichment comparison between consistently AD-specific and consistently control-specific NFKB1 ChIP-seq peaks. "Consistently specific" peaks were defined as those peaks that were AD- or control-specific in at least three case or control subjects, respectively. Results are shown for representative Cis-BP NF-κB motif, M05887_2.00. All results are listed in **S7** and **S9 Tables**.

peaks specific for controls (**Fig 5**). Importantly, there is AD specificity to the overlap of NFKB1 ChIP-seq peaks with AD-specific genes with AD-specific NFKB1 peaks overlapping more AD-specific genes (~80%) while control specific NFKB1 peaks overlapped only ~35% of AD-specific genes (**Fig 5**). These results are consistent with NFKB acting as both an activator and repressor depending on the context [25,26]. Collectively, these data indicate strong

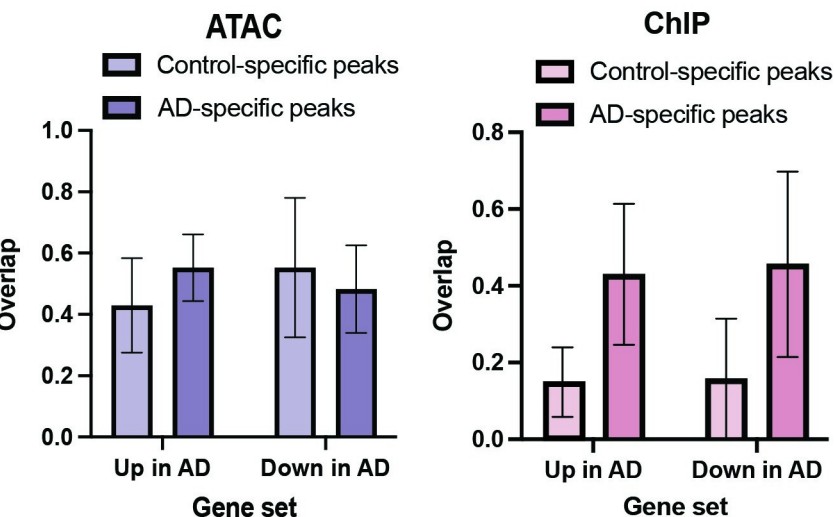

**Fig 5. Overlap of genes expressed in an AD-dependent manner with Control-specific and AD-specific ATAC-seq and ChIP-seq peaks.** AD-dependent genes can be upregulated in AD (Up in AD) or downregulated in AD (Down in AD). Overlap with of these gene sets with Control-specfic and AD-specific ATAC and ChIP-seq peaks is shown.

agreement between AD- and control-specific gene expression, chromatin accessibility, and NFKB1 binding.

## Allele-dependent chromatin accessibility at AD risk loci

Accumulating evidence reveals the important roles of allele-dependent gene regulatory mechanisms in various diseases [21,27–29]. To identify such mechanisms, we performed whole-genome sequencing of all subjects to identify the alleles present in AD genetic risk variants (**S11 Table**). We integrated these data with the functional genomics data produced in this study using the MARIO method, which measures the allele dependence of sequencing reads at genetic variants that are heterozygous [21]. Collectively, there were an average of 2.3 (range 0–5) heterozygous loci in AD cases and 2.6 (range 0–6) heterozygous loci among the controls (**S11 Table**), providing 124 opportunities to discover allele-dependent ATAC-seq or NFKB1 peaks at AD genetic risk variants.

Sixty AD-associated variants were located within an ATAC-seq peak in at least one subject and were found to be heterozygous in that subject. Notably, 36 of these 60 (60%) variants produced allele-dependent ATAC-seq peaks (**S12 Table** and **Fig 6**). 24 AD risk variants in 17 haplotypes overlap ATAC-peaks consistently found in subjects with AD. Of these, five overlap ATAC-seq peaks consistently found in subjects with AD but not consistently found in controls. Collectively, AD risk variants with allelic chromatin accessibility were found at nine independent risk loci (31% of AD risk loci). In 16 of the AD risk variants that were heterozygous and overlapped with ATAC-seq peaks, we observed allelic imbalance in multiple subjects. For example, at rs10791824 near the ovo-like transcriptional repressor 1 (*OVOL1*) gene, we measured a strong preference for the A allele across six individuals, with 85–100% of reads for all subjects having the A allele (total of 112 vs. 10, A vs. T reads). Twenty-eight AD risk variants with allelic ATAC-seq peaks were found to be eQTLs in stimulated CD4$^+$ T cells based on DICE, as curated by the eQTL catalogue [30] (**S13 Table**); however, these associations of allelic expression were not robust enough to remain after multiple testing correction after accounting for all of the eQTL measurements in that study (i.e., across many cell types with and without stimulation).

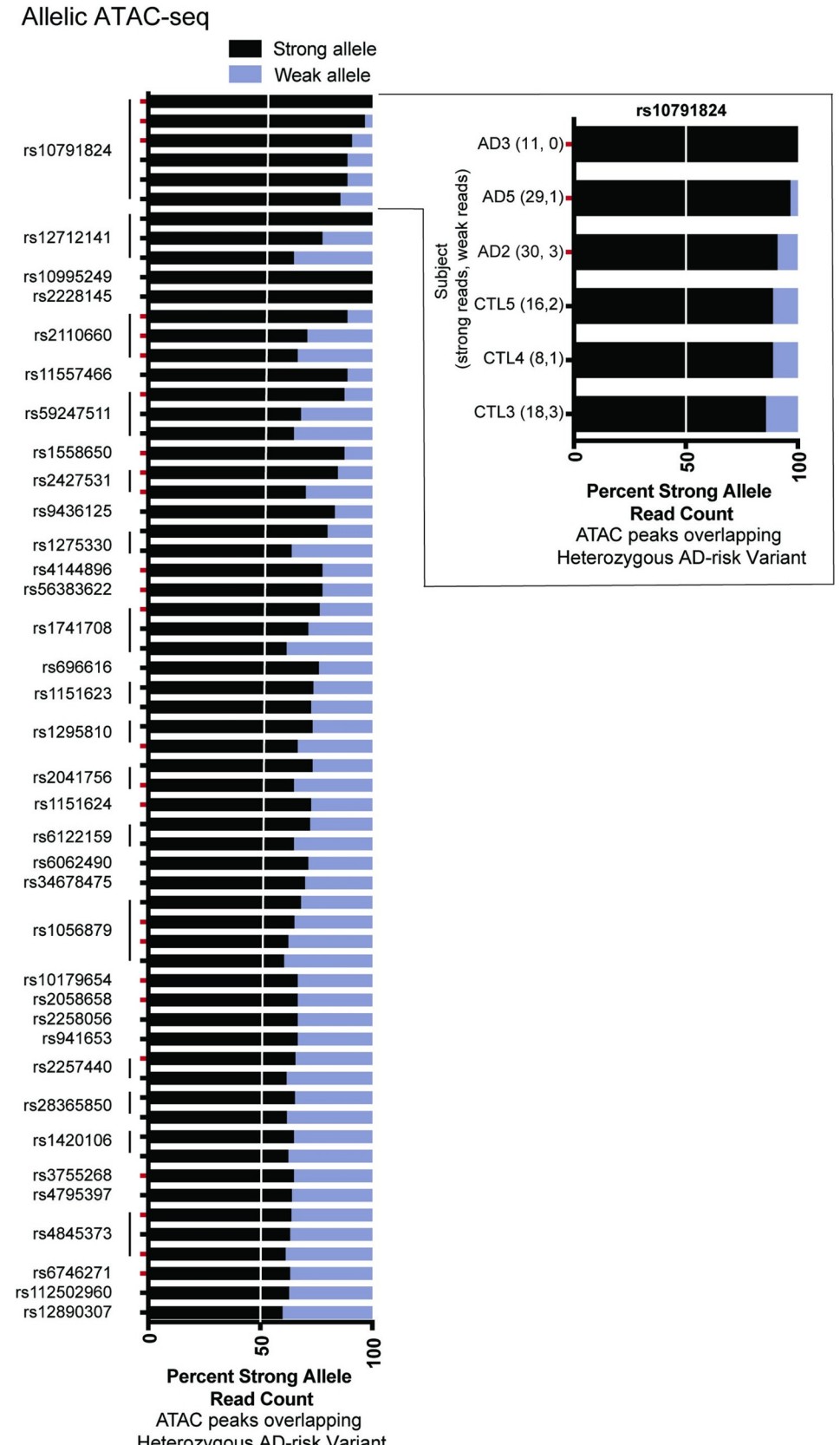

**Fig 6. Allele-dependent chromatin accessibility at AD risk loci.** (A) AD-associated genetic risk variants with allele-dependent ATAC-seq peaks in CD4$^+$ T cells. Each variant is heterozygous and located within an ATAC-seq peak in the indicated individual, facilitating measurement of allelic ratios informatics operator (MARIO) analysis to identify allele-dependent behavior. Red tick marks indicate allelic accessibility in a subject who with AD. Complete data are presented in S12 Table. All results shown have MARIO allelic reproducibility score (ARS) values $\geq$ 0.4, and hence, are allele-dependent. In the cutout, the participant identifier and reads under the ATAC-seq peak overlapping rs10791824 mapping to the strong and weak bases are provided.

In contrast to the large number of observed allelic ATAC-seq peaks, only six unique AD-associated variants were located within at least one NFKB1 ChIP-seq peak and also heterozygous in any of the 12 subjects; moreover, none of them demonstrated genotype-dependent activity. Hence, future studies examining larger cohorts are required to identify allelic NFKB1 binding activity.

## Discussion

Taken together, our data support a model in which stimulated peripheral blood CD4$^+$ T cells from patients with active AD exhibit extensive differential chromatin accessibility and NFKB1 binding compared to their matched controls. We identified genotype-dependent chromatin accessibility for 36 AD genetic variants. Collectively, this study highlights plausible genetic risk mechanisms for AD via disease-specific epigenetic factors enriched at AD genetic risk loci. Current databases of eQTLs are not specific to patients with AD or other allergic diseases, and do not contain a sufficient number of participants to have the statistical power to allow the identification of moderate-sized eQTLs. Therefore, it is important to continue collecting control and AD-specific eQTL datasets. It is also important to perform molecular studies assessing genotype-dependent regulatory activity of AD risk variants in the context of CD4$^+$ T cells to definitively identify allelic transcriptional mechanisms at these loci.

This study had limitations. We used strong stimulation (CD3/CD28 cross-linking) to model immune activation in patients. Use of different levels of stimulation may reveal additional disease-specific differences in future studies. We used CD4$^+$ T cells in our functional genomic assays to quantitatively assess disease-specific and genotype-dependent mechanisms. The disease- and genotype-dependent effects observed in this study can be refined and validated in specific immune cell subsets as the technology and analytical framework for quantitative comparisons at the single-cell level are refined. This study used a paired analysis of cases with demographically matched controls. The findings of this manuscript represent epigenetic differences that are consistent across these matched pairs. Future studies will be needed in which ancestry and age are carefully matched for all cases and controls to support a pooled analytical strategy. This study used a paired analysis of cases with demographically matched controls. The findings of this manuscript represent epigenetic differences that are consistent across these matched pairs. Future studies will be needed in which age is carefully matched for all cases and controls to support a pooled analytical strategy. The paucity of AD-associated NFKB1 binding variants could in large part be due to the lower number of NFKB1 ChIP-seq peaks.

Differences in the subtypes of CD4$^+$ T cells in the peripheral blood of patients with moderate-to-severe AD might explain some of the differential functional genomic effects identified in this study. AD patients exhibit an expansion of the T helper (Th)-9 subset of CD4$^+$ T cells and an increased frequency of circulating CD25hiFoxp3+ T cells compared to controls [31–33]. Skin-homing Th22 T cell levels are also increased in patients with AD across all ages [32]. Additionally, the T cell profile naturally changes with age, and patients with AD display a unique profile of changes [34]. Natural aging leads to both quantitative and qualitative changes

in the CD4$^+$ T-cell compartment [35]. Although our study did not differentiate between the different CD4$^+$ T cell subtypes, our case-control comparisons were performed between demographically matched controls. Future studies should investigate the role of these different subtypes based on the age and prevalence of genotype-dependent transcriptional dysregulation in AD patients compared to controls.

This study is important for the understanding of atopic dermatitis because a mechanistic understanding of the etiology of the disease could enable efficacious predictive tools and preventative therapeutics. Because atopic dermatitis is the first diagnosis of most patients who find themselves on the atopic march from AD to food allergy to asthma to allergic rhinitis, strategies to prevent and halt AD have the potential to prevent other atopic diseases. Current treatments for AD include the use of topical steroids and anti-inflammatory non-steroidal agents, such as anti-type 2 immunity biological agents and Janus kinase inhibitors, as well as aggressive moisturization and systemic immunosuppression in severe cases. There is an urgent need for novel therapeutics to address the complexity of this disease and to contribute to the development of personalized medicine against various subtypes/phenotypes that differ by age, disease chronicity, and underlying molecular mechanisms [4,36]. The results of this study support a continued focus on the development of therapeutics aimed at inhibiting the NF-κB signaling pathway. Numerous studies in mice and humans provide further impetus to elucidate this mechanism in AD [37,38].

In conclusion, the results of this study support a model in which stimulated CD4$^+$ T cells from patients with AD display disease- and allele-dependent differences in chromatin accessibility and disease-dependent differences in NFKB1 binding. Based on the broad genotype-dependent chromatin accessibility at AD risk variants in stimulated CD4$^+$ T cells, our data support allelic transcriptional regulation as an important epigenetic mechanism mediating disease risk in AD.

## Methods

### Ethics statement

This study was approved by the Cincinnati Children's Hospital Institutional Regulatory Board (IRB #2018–4254). Prior to participation, all adults, children over 13 years of age, and parents of children below 13 years of age provided written informed consent; children 13 years and older provided assent for their participation.

**Collection of AD-associated genetic risk variants.** One hundred and twenty-two genetic variants that reached genome-wide significance in a GWAS of AD were identified from the Genome Wide Association Catalogue (https://www.ebi.ac.uk/gwas/) [39] and a genetic association study using the Illumina ImmunoChiP [40]. Twenty-nine independent genetic risk loci were identified via linkage disequilibrium pruning ($r^2 < 0.2$). We identified 3,143 AD risk variants across these 29 loci by accounting for linkage disequilibrium ($r^2 > 0.8$) based on 1000 Genomes Data [41] in the ancestry(ies) of the initial genetic association using PLINK (v.1.90b) [42]. S1 Table presents the twenty-nine AD risk loci that reached genome-wide significance at the beginning of this study; this supplemental table includes information regarding the major, minor, reference, non-reference, and risk alleles as well as allele frequencies, disease odds ratios, and additional associated diseases and phenotypes.

**Patient recruitment.** Patients with moderate-to-severe AD were recruited for this study from the Cincinnati Children's Hospital Allergy Clinics, the Bernstein Allergy Group, and the Bernstein Clinical Research Center.

Matched healthy controls were recruited via advertisements. To reduce heterogeneity, the study inclusion criteria for subjects with AD were: 1) presence of atopy established by positive

aeroallergen skin prick test and/or elevated serum total IgE levels; and 2) moderate-to-severe AD defined as an EASI score $\geq$ 17 and investigator global assessment score $\geq$ 3 [43]. These AD severity tools have previously been validated for AD severity scoring [43,44]. Control subjects were included if they had no history of atopic disease and displayed negative results from aeroallergen skin prick testing performed at their enrollment visit. Exclusion criteria included a history of any biologic therapy, oral steroids, or immunosuppressive medications in the past 6 months due to their effect on T cells and the transcriptome. **S13 Table** provides the age, sex, self-reported race and ethnicity of each subject in this study.

**CD4$^+$ T cell stimulation.**   We isolated the peripheral blood mononuclear cells (PBMCs) from AD and control individuals using Ficoll-Paque (GE Healthcare) density gradient separation. Then, 51.5 ± 20.1 million PBMCs were isolated from each participant. Whole genome sequencing of the DNA extracted from PBMCs was performed to identify the genetic variants. CD4$^+$ T cells were isolated using magnetic column separation (Miltenyi Biotec; CD4$^+$ T cell isolation kit, human); 17.7 ± 7.2 million CD4+ T cells were isolated from each subject. To activate NF-κB, we stimulated CD4$^+$ T cells with CD3/CD28 crosslinking for 45 h (Gibco; Dynabeads Human T-Activator CD3/CD28). We performed ChIP-seq (2 million stimulated CD4$^+$ T cells), ATAC-seq (50,000 stimulated CD4$^+$ T cells), and RNA-seq assays (2.5 million stimulated CD4$^+$ T cells). The detailed methods for each genetic and genomic assay are provided below.

**ATAC-seq.**   Transposase Tn5 with adapter sequences was used to cut the accessible DNA [45]. These accessible DNA sequences with adaptor sequences were isolated, and libraries were prepared from 50,000 stimulated CD4$^+$ T cells using the OMNI ATAC protocol [46]. The libraries were sequenced at 150 bases per end on an Illumina NovaSeq 6000 at the Cincinnati Children's Hospital Medical Center (CCHMC) DNA Sequencing and Genotyping Core Facility. The quality of the sequencing reads was verified using FastQC (v. 0.11.2) (http://www.bioinformatics.babraham.ac.uk/projects/fastqc), and adapter sequences were removed using Cutadapt (Trimgalore v. 0.4.2). ATAC-seq reads were aligned to the human genome (hg19) using Bowtie2 [47]. Aligned reads were then sorted using samtools (v.1.8) [48] and duplicate reads were removed using Picard (v. 1.89) (https://broadinstitute.github.io/picard/). Peaks were called using MACS2 (macs2 callpeak -g hs -q 0.01) [49]. ENCODE blacklist regions (https://github.com/Boyle-Lab/Blacklist/tree/master/lists/hg19-blacklist.v2.bed.gz) [50] were removed. Differential chromatin accessibility was calculated using the MAnorm program [24] with thresholds of fold change greater than 1.5 and p-value less than 0.05.

**ChIP-seq.**   CD4$^+$ T cells from subjects were crosslinked, and nuclei were sonicated. Cells were incubated in a crosslinking solution (1% formaldehyde, 5 mM 4-(2-hydroxyethyl)-1-piperazineëthanesulfonic acid (HEPES) [pH 8.0], 10 mM sodium chloride, 0.1 mM ethylenediaminetetraacetic acid (EDTA), and 0.05 mM ethylene glycol tetraacetic acid (EGTA) in Roswell Park Memorial Institute (RPMI) culture medium with 10% fetal bovine serum (FBS)) and placed on a tube rotator at room temperature for 10 min. To stop crosslinking, glycine was added to a final concentration of 0.125 M, and the tubes were rotated at room temperature for 5 min. Cells were washed twice with ice-cold phosphate-buffered saline (PBS), resuspended in lysis buffer 1 (50 mM HEPES [pH 8.0], 140 mM NaCl, 1 mM EDTA, 10% glycerol, 0.25% Triton X-100, and 0.5% NP-40), and incubated for 10 min on ice. Nuclei were harvested after centrifugation at 5,000 rpm for 10 min, resuspended in lysis buffer 2 (10 mM Tris-HCl [pH 8.0], 1 mM EDTA, 200 mM NaCl, and 0.5 mM EGTA), and incubated at room temperature for 10 min. Protease and phosphatase inhibitors were added to both lysis buffers. Nuclei were resuspended in sonication buffer (10 mM Tris [pH 8.0], 1 mM EDTA, and 0.1% sodium dodecyl sulfate (SDS)). An S220 focused ultrasonicator (COVARIS) was used to shear chromatin (150–500-bp fragments) with 10% duty cycle, 175 peak power, and 200 bursts per cycle for 7 min. A portion

of the sonicated chromatin was run on an agarose gel to verify fragment sizes. Sheared chromatin was pre-cleared with 10 μL of Protein G Dynabeads (Life Technologies) at 4˚C for 1 h.

Immunoprecipitation of NFKB1-chromatin complexes was performed using an SX-8X IP-STAR compact automated system (Diagenode). Beads conjugated to antibodies against NFKB1 (Cell Signaling (D7H5M) Rabbit mAb #12540) were incubated with pre-cleared chromatin at 4˚C for 8 h. The beads were then washed sequentially with buffer 1 (50 mM Tris-HCl [pH 7.5], 150 mM NaCl, 1 mM EDTA, 0.1% SDS, 0.1% NaDOC, and 1% Triton X-100), buffer 2 (50 mM Tris-HCl [pH 7.5], 250 mM NaCl, 1 mM EDTA, 0.1% SDS, 0.1% NaDOC, and 1% Triton X-100), buffer 3 (2 mM EDTA, 50 mM Tris-HCl [pH 7.5] and 0.2% sarkosyl sodium salt), and buffer 4 (10 mM Tris-HCl [pH 7.5], 1 mM EDTA, and 0.2% Triton X-100). Finally, the beads were resuspended in 10 mM Tris-HCl (pH 7.5) and used to prepare the libraries via ChIPmentation [51].

The ChIP-seq libraries were sequenced as single-end 100-base reads on an Illumina Nova-Seq 6000 at the CCHMC DNA Sequencing and Genotyping Core Facility. The reads were processed and analyzed as described above for ATAC-seq. We also used publicly available NFKB1 ChIP-seq datasets (GSE126505) that were processed using the same analytical pipeline.

**RNA-seq.** Total RNA was extracted using the mirVANA Isolation Kit (Ambion) from stimulated CD4$^+$ T cells of the control and AD subjects 45 h post-stimulation. RNA-seq libraries were sequenced as paired ends with 150 bases. FastQC and cutadapt were used to verify the read quality and remove adapters, respectively, as described above. RNA-seq reads were aligned to the hg19 (GrCh37) genome build (National Center for Biotechnology Information, NCBI) using Spliced Transcript Alignment with a Reference (STAR, v. 2.5.2a) [52]. The program featureCounts (subread/1.6.2) was used to count the reads mapped to each gene [53]. The fragments per kilobase per million mapped fragments (FPKM) values for the relative expression of each gene were used to calculate the pairwise AD case/control fold-change. Differential expression for pairwise subject comparisons was established as a fold-change greater than 1.5.

**Whole genome sequencing and variant calling.** DNA was isolated using a PureLink Genomic DNA Kit (Thermo Fisher). Whole-genome sequencing was performed using DNBseq next-generation sequencing technology. Libraries were sequenced on an Illumina NovaSeq to generate 100-base paired-end reads. Sequencing reads were aligned, and variants were identified using the Genome Analysis Toolkit (GATK) Unified Genotyper following GATK Best Practices 3.3 [54–56].

**Enrichment analysis for functional genomic datasets.** We used the RELI algorithm to estimate the significance of the overlap between genomic features generated in this study (TF binding events, histone marks, ATAC-seq peaks, etc.) [21,57,58]. As input, RELI takes the genomic coordinates of peaks from two datasets. RELI then systematically intersects these coordinates with one another and the number of input regions overlapping the peaks is counted. Next, a p-value describing the significance of this overlap is estimated using a simulation-based procedure in which the peaks from the first dataset are randomly distributed within the union coordinates of open chromatin from human cells. A distribution of expected overlap values is then created from 2,000 iterations of randomly sampling from the negative set, each time choosing a set of negative examples that match the input set in terms of the total number of genomic loci. The distribution of the expected overlap values from the randomized data resembles a normal distribution and can thus be used to generate a Z-score and corresponding p-value estimating the significance of the observed number of input regions that overlap each dataset.

**Identification of allelic ATAC-seq and ChIP-seq reads using MARIO.** To identify possible allele-dependent mechanisms in our functional genomics datasets, we applied our

MARIO method [21]. In cases where a given variant is heterozygous in the cell assayed, both alleles are available for the TF to bind or for the chromatin to be accessible or not, offering a natural control for one another since the only variable that has changed is the allele. In brief, MARIO identifies common genetic variants that are (1) heterozygous in the assayed cell line (using NGS DNA sequencing data) and (2) located within a peak in a given ChIP-seq or ATAC-seq dataset. MARIO then examines the sequencing reads that map to each heterozygote within each peak for imbalance between the two alleles. To estimate the significance of the degree of allelic imbalance of a given dataset at a given heterozygote, we developed the MARIO Allelic Reproducibility Score (ARS), which is based on a combination of two *predictive variables*: the total number of reads overlapping the variant and the imbalance between the number of reads for each allele. We report allelic accessibility and NFKB1 binding at AD genetic risk variants in our ATAC-seq data with an ARS greater than or equal to 0.4 which is considered significantly allelic [21].

## Supporting information

**S1 Fig. ATAC-seq and NFKB1 ChIP-seq peak pairwise normalized read count comparisons.** Each dot represents a peak that is present in either of the two samples. The X- and Y-axis indicate the number of normalized read counts for that peak in the two samples. The red line indicates peaks with similar normalized read counts between the two samples. **A**. ATAC- seq scatter plot analysis for controls. **B**. ATAC-seq scatter plot analysis for AD cases. **C**. ChIP-seq scatter plot analysis for controls. **D**. ChIP-seq scatter plot analysis for AD cases. E. Global Variability analysis of the ATAC and ChIP peaks across AD and controls.
(PDF)

**S2 Fig. Case/control pairwise differential ATAC-seq analysis and transcription factor motif enrichment.** (**Left plots**). For each of the six matched sample pairs, differential ATAC-seq peak analysis was performed. ATAC-seq peaks that are equally strong between case and controls ("shared"), significantly stronger in the control ("control-specific"), or significantly stronger in the case ("AD-specific") were identified using MAnorm (see Methods). The percent of peaks falling within each category is indicated to the left of each heatmap. Heatmap color indicates the normalized strength of the ATAC-seq signal. (**Right plots**). Transcription factor motif enrichment analysis was performed for each sample pair (see Methods). Each dot represents a transcription factor motif. The X-axis and Y-axis indicate the p-value of enrichment within control-specific and AD-specific ATAC-seq peaks, respectively. Motif family color key is provided at the top of each plot.
(PDF)

**S3 Fig. Number of consistently differential ATAC-seq and ChIP-seq peaks across sample pairs.** ATAC-seq and ChIP-seq peaks were called for all cases and controls. For each matched pair of samples, a MAnorm analysis identified shared peaks, control-specific peaks and AD-specific peaks (see Methods). Bar plots indicate the distributions of the number of control- specific (left) or case-specific peaks (right) for ATAC-seq (A,B) and ChIP-seq (C,D). For example, there are approximately 10 ATAC-seq peaks in the genome that are control-specific in 4 of the 6 matched pairs (see panel A).
(PDF)

**S4 Fig. Case/control pairwise differential NKFB1 ChIP-seq analysis and transcription factor motif enrichment.** (**Left plots**). For each of the six matched sample pairs, differential NKFB1 ChIP-seq peak analysis was performed. ChIP-seq peaks that are equally strong between case and controls ("shared"), significantly stronger in the control ("control-specific"),

or significantly stronger in the case ("AD-specific") were identified using MAnorm (see Methods). The percent of peaks falling within each category is indicated to the left of each heatmap. Heatmap color indicates the normalized strength of the ChIP-seq signal. (**Right plots**). Transcription factor motif enrichment analysis was performed for each sample pair (see Methods). Each dot represents a transcription factor motif. The X-axis and Y-axis indicate the p-value of enrichment within control- specific and AD-specific ChIP-seq peaks, respectively. Motif family color key is provided at the top of each plot.
(PDF)

**S5 Fig. Transcription factor motif enrichment comparison between consistently control-specific and consistently AD-specific NFKB1 ChIP-seq peaks.** The top five enriched TF motif families are shown. **A.** Percent of peaks containing predicted binding sites for the indicated motif. **B.** Motif enrichment p-value within those peaks (see Methods).
(PDF)

**S6 Fig. RNA-seq GO enrichment results.** Pairwise differential gene expression analysis was performed by comparing RNA-seq data for each of the cases with their matched controls (see Methods). Genes with 1.5-fold changes or greater in three or more pairs were identified. Pathway enrichment analysis was performed on these gene sets. The top five enriched GO biological pathways and GO molecular function categories are indicated within the top and bottom tables, respectively.
(PDF)

**S1 Table. Atopic Dermatitis risk variants and loci.**
(XLSX)

**S2 Table. Quality control summary.**
(XLSX)

**S3 Table. Regulatory Element Locus Intersection (RELI) analysis of ATAC-seq and ChIP datasets.**
(XLSX)

**S4 Table. Transcription factor motif analysis of ATAC-seq datasets.**
(XLSX)

**S5 Table. Transcription factor motif analysis of ATAC-seq peaks consistently found in samples from patients with atopic dermatitis and controls.**
(XLSX)

**S6 Table. Regulatory Element Locus Intersection (RELI) analysis of NFKB1 ChIP-seq datasets.**
(XLSX)

**S7 Table. Transcription factor motif analysis of NFKB1 ChIP-seq datasets.**
(XLSX)

**S8 Table. Regulatory Element Locus Intersection (RELI) integration of RNA-seq, ATAC-seq, and NFKB1 ChIP-seq datasets.**
(XLSX)

**S9 Table. Transcription factor motif analysis of NFKB1 ChIP-seq peaks consistently found in samples from patients with atopic dermatitis and controls.**
(XLSX)

**S10 Table. Analysis of RNA-seq datasets.**
(XLSX)

**S11 Table. Identification of AD risk variants in DNA-seq datasets.**
(XLSX)

**S12 Table. Identification of genotype-dependent.**
(XLSX)

**S13 Table. Demographics of participants in this study.**
(DOCX)

## Acknowledgments

We would like to thank the following physicians for their support in establishing our research clinic: Juan Pablo Abonia, MD; Jonathan Bernstein, MD; Sheharyar Durrani, MD; Stephanie Ward, MD; Justin Greiwe, MD; Michelle Lierl, MD; and Kimberly Risma, MD, PhD. We appreciate the support Bahram Namjou, MD and Andrew VonHandorf, PhD. Some figures were created in BioRender.

## Author Contributions

**Conceptualization:** Amy A. Eapen, Ashley L. Devonshire, Leah C. Kottyan.

**Data curation:** Sreeja Parameswaran, Lee E. Edsall, Omer Donmez, Xiaoming Lu, Marissa Granitto, Matthew T. Weirauch.

**Formal analysis:** Sreeja Parameswaran, Lee E. Edsall, Xiaoming Lu, Marissa Granitto, Xiaoting Chen, Kenneth Kaufman, Matthew T. Weirauch, Leah C. Kottyan.

**Funding acquisition:** Marc E. Rothenberg, Matthew T. Weirauch, Leah C. Kottyan.

**Investigation:** Amy A. Eapen, Sreeja Parameswaran, Carmy Forney, Lee E. Edsall, Daniel Miller, Omer Donmez, Katelyn Dunn, Hope Rowden, Adam Z. Magier, Mario Pujato, Xiaoting Chen, David I. Bernstein, Ashley L. Devonshire, Matthew T. Weirauch, Leah C. Kottyan.

**Methodology:** Carmy Forney, Omer Donmez, Xiaoting Chen, Matthew T. Weirauch, Leah C. Kottyan.

**Project administration:** Amy A. Eapen, Leah C. Kottyan.

**Resources:** David I. Bernstein, Marc E. Rothenberg, Matthew T. Weirauch, Leah C. Kottyan.

**Software:** Sreeja Parameswaran, Mario Pujato, Xiaoting Chen, Matthew T. Weirauch.

**Supervision:** Matthew T. Weirauch, Leah C. Kottyan.

**Visualization:** Sreeja Parameswaran, Leah C. Kottyan.

**Writing – original draft:** Amy A. Eapen, Ashley L. Devonshire, Marc E. Rothenberg, Matthew T. Weirauch, Leah C. Kottyan.

**Writing – review & editing:** Amy A. Eapen, Carmy Forney, Lee E. Edsall, Daniel Miller, Omer Donmez, Katelyn Dunn, Xiaoming Lu, Marissa Granitto, Hope Rowden, Adam Z. Magier, Mario Pujato, Xiaoting Chen, Kenneth Kaufman, David I. Bernstein, Ashley L. Devonshire, Marc E. Rothenberg, Matthew T. Weirauch, Leah C. Kottyan.

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
