## [Decision Letter · Decision Letter 0]

20 Dec 2021

Dear Dr Kottyan,

Thank you very much for submitting your Research Article entitled 'Epigenetic and Transcriptional Dysregulation in CD4+ T cells of Patients with Atopic Dermatitis' to PLOS Genetics.

The manuscript was fully evaluated at the editorial level and by independent peer reviewers. The reviewers appreciated the attention to an important problem, but raised some substantial concerns about the current manuscript. Based on the reviews, we will not be able to accept this version of the manuscript, but we would be willing to review a much-revised version. We cannot, of course, promise publication at that time.

If you decide to revise the manuscript for further consideration at PLOS Genetics, please aim to resubmit within the next 60 days, unless it will take extra time to address the concerns of the reviewers, in which case we would appreciate an expected resubmission date by email to plosgenetics@plos.org.

[LINK]

We are sorry that we cannot be more positive about your manuscript at this stage. Please do not hesitate to contact us if you have any concerns or questions.

Yours sincerely,

Devin M Absher

Associate Editor

PLOS Genetics

John Greally

Section Editor: Epigenetics

PLOS Genetics

Reviewer's Responses to Questions

**Comments to the Authors:**

Reviewer #1: Thanks to the authors for this paper. I have only two criticisms:

1. The paper largely focuses on overlaps in genomic regions without applying any statistical evidence that those overlaps are more extreme than would be expected by chance alone. In the resubmission I would advise including hypergeometric testing or something similar to asses the strength of evidence.

2. The statement below is copied from the manuscript:

The 100 kB region of DNA around AD-specific gene sets widely overlapped (94.7-100%) the ATAC-seq peaks in the six subjects with AD (Supplemental Table 8). There was substantial overlap (26.3-68.4%) between the 100 kb region of DNA around ADspecific gene sets with the AD-specific ATAC-peaks, indicating that possible enhancers proximal to the AD-specific genes were accessible for transcription in an AD-specific manner.

The above gives no indication of the directionality of effect. Can you produce a graph or something similar to show the relationship between chromatin accessibilty and gene expression i.e logFC case v control (GE) v logFC case v control (ATAC)

3. Likewise is it possible to include a figure demonstrating the allele-specific chromatin accessibility i.e ATAC reads stratified by genotype in cases and controls or something similar?

Thanks

Reviewer #2: This is an interesting report from Eapen et al., that characterizes the epigenetic and transcriptional dysregulation of CD4 T cell in patients with atopic dermatitis (AD). AD affects approximately 20% of children and high rate of persistence into adulthood. There are 29 independent risk haplotypes identified. It is well know that CD4 T cells are the major effector cell type for AD and that NFKB signaling mediates the pathogenic inflammation. This paper seeks to determine if there are upstream effects in CD4 T cells at the level of epigenome and transcriptome changes that facilitate pathogenic T cell responses. The authors use a combination of genomic techniques to determine if AD risk haplotypes demonstrate altered chromatin accessibility, NFKB binding and gene expression changes consistent with AD using a case/control study design in CD4 T cells in patients with AD. They demonstrate that in stimulated CD4 T cells taken from patients with AD that open chromatin regions were enriched for AD risk variants and that there was strong enrichment for NFKB binding motifs in these peaks in AD patients but not in controls. They also demonstrated over 60 instances of genotype-dependent chromatin accessibility for 36 AD risk variants. Together they conclude that allele specific epigenetic and transcriptional regulation is an important feature of CD4 T cell responses to stimulation in patients with AD. While these results are interesting, there are several significant issues that need to be addressed to improve the overall quality of this manuscript. These are listed below in no specific order of importance:

1. The text is missing some basic proofreading

2. Testing the hypothesis that “AD loci may be epigenetically regulated” is vague (pp. 2, 128).

3. The FRiP scores are quite low with average of 32% indicating that the ATAC-seq data may have QC issues (typically >50% is expected), please address.

4. Analysis of shared peaks was performed by using pairwise assessments. It is not clear how case/control pairs were selected and why a more exhaustive case/control analysis was not performed (i.e., pairwise comparisons using all possible pairs), or why consensus peaks were not called from pooled data.

5. More generally, how were subjects matched?

6. After the pairing approach used to identify AD- or control-specific ATAC peaks, peaks that were AD- or control-specific in 3 or more of the pairings were classified as “consistently AD/control” (pp. 5, 176-178). It is not stated how much these peak sets overlap each other. Based on the data in the supplement (Supplemental Table 3), they share 80% of LD SNPs so it is likely they have some overlap.

7. Based on shared LD and Tag SNPs (Supplemental Table 3) it is NOT clear that chromatin is accessible in a “disease specific manner” (pp. 5, 181-182). Twelve of the 13 AD risk loci overlapped by “consistently AD” peaks are also overlapped by “consistently control” peaks.

8. On pp. 5, 170-171, authors state 75-88.4% of ATAC peaks were shared between AD and demographically matched controls, but on pp. 8, 231 authors state a median of 91.9% of ATAC peaks were shared. It is unclear how this inconsistency arose or if the comparison being made is different in the second statement in a way that is not clear from the text.

9. On pp. 22, 471-472 in the METHODS section, authors state that reads for ChIP-seq were processed the same way as ATAC-seq reads, but the MACS2 parameters needed to properly call peaks in these two data types should be different to avoid errors in the genomic coordinates of the called peaks, especially excluding parameters ‘-nomodel --shift --extsize’ for ATAC-seq peak calling.

10. On pp. 8, 228-231, it states that there is more variability in overlap between subject matched pairs for ChIP vs. ATAC peaks, but this may be entirely due to the fact that there is substantial variability in the number of ChIP peaks across samples (Supplemental Table 2). It would be useful to compare variability of overlap within controls and within AD to see if the variability between control vs. AD is any greater than the large variability already present in the samples.

11. The biological processes associated with the differentially expressed genes (15 are expressed 1.5x higher in AD, 16 are expressed 1.5x lower in AD; shown in a table in Supplemental Figure 6) are not particularly compelling. These seem like very high-level and general processes and many of the enriched pathways include hundreds or even over 1000 genes. More generally, only finding 31 differentially expressed genes seems surprisingly low.

12. The case/control RNA-seq comparisons were only performed pairwise (pp. 11, 280), which seems like a missed opportunity for comparing pooled case vs. pooled control data, which would presumably have more power to detect differentially expressed variants (this applies to the pairwise comparisons used more generally)

13. On pp. 12, 289-299, authors describe how a large % of AD-specific ATAC peaks overlap the 100kb region around AD-specific gene sets. They then state that NFKB1 ChIP-seq peaks overlap a large % of the TSS of AD-specific genes (they don’t state whether it is AD-specific NFKB1 peaks or ALL peaks). They then state that a lower % of AD-specific NFKB1 peaks overlap the AD-specific genes (does this mean that the peaks overlap the transcripts?). It would be helpful to show additional comparisons here to make the argument that control- and AD-ChIP/ATAC peaks are truly specific for control- and AD-genes. For example, what % of AD-specific genes show control-specific ATAC peaks in the 100kb vicinity? Is it similar to the % of AD-specific ATAC peaks, or is the % of AD-specific ATAC peaks significantly higher in the vicinity of AD-specific genes, etc.

14. Authors used MARIO method to integrate information across ATAC/ChIP/RNA-seq data with WGS for each of their subjects and find allele dependence of sequencing reads (read depth?) at het variants (het across the sample, or within individuals?) (pp. 13, 304-307)

15. Pp. 13, 311-313: The authors state that the MARIO analysis discovers AD-associated variants in ATAC peaks that are het in some subjects and produce allele-dependent ATAC peaks. This seems like a circular argument: the MARIO analysis uses differences in sequencing reads combined with genotyping data to discover alleles that affect sequencing reads, so why is it striking that it finds alleles that produce allele dependent ATAC peaks, since that is the very signal it is based on?

16. Fewer AD-associated NFKB1 binding variants were found, but this may be largely driven by the fact that there were ~7x more ATAC peaks than ChIP peaks (as the authors acknowledge pp. 14, 328-329).

17. It would be interesting search the AD-associated alleles in large public datasets and look analyze their frequencies, whether they have been associated with other diseases, etc.

18. The author should make an effort in the discussion, to connect how their data will impact the “atopic march” mentioned in the introduction.

19. There is an error in the readme file tab in Table 12 that should be changed. See line 15 of the table. S_read mislabeled as weak reads.

**Have all data underlying the figures and results presented in the manuscript been provided?**

Reviewer #1: Yes

Reviewer #2: Yes

PLOS authors have the option to publish the peer review history of their article (what does this mean?). If published, this will include your full peer review and any attached files.

Reviewer #1: **Yes: **David Martino

Reviewer #2: No

---

## [Decision Letter · Decision Letter 1]

20 Apr 2022

Dear Dr Kottyan,

We are pleased to inform you that your manuscript entitled "Epigenetic and Transcriptional Dysregulation in CD4+ T cells in Patients with Atopic Dermatitis" has been editorially accepted for publication in PLOS Genetics. Congratulations!

Yours sincerely,

Devin M Absher

Associate Editor

PLOS Genetics

John Greally

Section Editor: Epigenetics

PLOS Genetics

Comments from the reviewers (if applicable):

Reviewer's Responses to Questions

**Comments to the Authors:**

Reviewer #1: My comments were addressed, thanks to the authors.

Reviewer #2: The authors have adequately addressed the concerns raised in the original version of the manuscript

**Have all data underlying the figures and results presented in the manuscript been provided?**

Reviewer #1: Yes

Reviewer #2: Yes

PLOS authors have the option to publish the peer review history of their article (what does this mean?). If published, this will include your full peer review and any attached files.

Reviewer #1: No

Reviewer #2: No

**Data Deposition**

http://datadryad.org/submit?journalID=pgenetics&manu=PGENETICS-D-21-01587R1

**Press Queries**

---

## [Editor Report · Acceptance letter]

11 May 2022

PGENETICS-D-21-01587R1 

Epigenetic and Transcriptional Dysregulation in CD4+ T cells in Patients with Atopic Dermatitis 

Dear Dr Kottyan, 

We are pleased to inform you that your manuscript entitled "Epigenetic and Transcriptional Dysregulation in CD4+ T cells in Patients with Atopic Dermatitis" has been formally accepted for publication in PLOS Genetics! Your manuscript is now with our production department and you will be notified of the publication date in due course.

With kind regards,

Agnes Pap

PLOS Genetics

On behalf of:
